# Transitioning Bodies. The Case of Self-Prescribing Sexual Hormones in Gender Affirmation in Individuals Attending Psychiatric Services

**DOI:** 10.3390/brainsci8050088

**Published:** 2018-05-14

**Authors:** Antonio Metastasio, Attilio Negri, Giovanni Martinotti, Ornella Corazza

**Affiliations:** 1Camden and Islington NHS Foundation Trust, London NW1 0PE, UK; 2Centre for Clinical & Health Research Services, School of Life and Medical Sciences, University of Hertfordshire, Hatfield AL10 9AB, UK; ngrttl@gmail.com (A.N.); giovanni.martinotti@gmail.com (G.M.); o.corazza@herts.ac.uk (O.C.); 3Department of Neuroscience, Imaging, and Clinical Science, “G. d’Annunzio” University of Chieti-Pescara, 66100 Chieti, Italy

**Keywords:** transgender, gender reassignment, gender affirmation, self-medication, hormonal replacement therapy (HRT), LGTBQ health, gender dysphoria, do it yoursfelf (DIY), identity, barriers to care, discrimination

## Abstract

Self-prescribing of sexual hormones for gender affirmation is a potentially widespread and poorly studied phenomenon that many clinicians are unaware of. The uncontrolled use of hormones poses significant health hazards, which have not been previously reported in the literature. We have collected seven clinical cases in general adult psychiatry settings (both inpatient and outpatients), describing transgender and gender non-conforming individuals’ (TGNC) self-prescribing and self-administering hormones bought from the Internet without any medical consultation. Among these cases, two were taking androgens, and the rest were taking oestrogens. The main reason for self-administration of hormones seems to be the lack of access to specialised care due to discrimination and long waiting lists. We advocate for clinicians to be aware of the phenomenon and proactively help TGNC individuals by enquiring about self-prescribing of hormones, providing information and referring to the most appropriate treatment centre as well as encourage a public debate on the discrimination and the stigma that TGNC population suffer from. Overall, there is an urgent need for the implementation of different and innovative health care services for TGNC individuals as well as more targeted prevention strategies on such underreported and highly risky behaviours. Furthermore, it is necessary for every clinician involved in the care for TGNC people to be aware of their special needs and be able to be an allied and an advocate to help in reducing stigma and discrimination that affect the access to care for this often underserved population.

## 1. Introduction

A significant proportion of the population defines themselves as transgender, intersex non-binary or gender non-conforming (in this paper, we will use “TGNC” for “transgender and gender nonconforming” people as recommended by the American Psychological Association (APA) guidelines [1]). A recent demographic study estimates that in the USA 0.39% (about one million people) of the population define themselves as TGNC [2]. The exact number, however, might be bigger, and it is very difficult to quantify the precise number due to the complex methodology in estimating the numbers when gender non-conforming individuals are also considered in the statistics [3]. To estimate the same data in the UK, it is more difficult because the Office for National Statistics (ONS) does not produce estimates of the number of TGNC people living in the United Kingdom (Office for National Statistics [4]). The Gender Identity Research and Education Society (GIRES) estimates a prevalence of 1% TGNC individuals in the UK adult population [5]. TGNC people are individuals that do not identify or exclusively identify with the sex assigned to them at birth. Intersex individuals are people with a less common combination of hormones, chromosomes, and anatomy that are used to assign sex at birth. There are many examples such as Klinefelter Syndrome, Androgen Insensitivity Syndrome, and Congenital Adrenal Hyperplasia. Non Binary people are individuals that do not identify themselves completely as female/male or woman/man. Gender non-conforming people are, according to the American Psychological Association “those who have a gender identity that is not fully aligned with their sex assigned at birth” [1].

A significant number of these individuals often decide to undergo a gender affirmation process. This process consists of using sexual hormones and often undergoing surgery to affirm to the gender that they belong to. In the UK, the prevalence of the population that has sought medical care is estimated to be 0.025%, and about 0.015% are likely to have undergone a transition [5].

A recent survey made by the University of California, Los Angeles (UCLA) Centre for Health Policy and Research showed that 27% of youth between 12 and 17 in California are gender non-conforming [6]. According to another document from the same institution, such a population typically presents a “*conflict between a person’s physical or assigned gender and the gender with which he/she/they identify. People with gender dysphoria may be very uncomfortable with the gender they were assigned, sometimes described as being uncomfortable with their body (particularly developments during puberty) or being uncomfortable with the expected roles of their assigned gender*” [7].

It could be, therefore, be argued that gender affirmation is a very important procedure to improve the quality of life and mental wellbeing of TGNC individuals. However, very little attention has been paid to this phenomenon.

A recent prospective study [8] assessing the psychopathology during the gender affirmation process has shown that the psychoneurotic distress measured with the Symptom Checklist-90 Revised SCL-90-R, improves after the start of the hormonal treatment, and anxiety, depression, interpersonal sensitivity, and hostility also tend to improve. This progress is so important that after the completion of the gender affirmation procedure via hormonal treatment and surgery, the psychopathology (assessed with a specific scale) is comparable to the one of the general population [8]. Another prospective study has also demonstrated that people with gender dysphoria treated with hormones presented a significant improvement at the Body Uneasiness Test (BUT) compared to the non-treated condition [9].

The use of hormones, however, might have significant side-effects or may lead to severe medical complications [10]. In particular, Cross-sex Hormone Therapy (CHT) female to male has been associated with a potential risk of cardiovascular disease, cancer (breast, ovarian and endometrial), osteoporosis; in the case of male to female therapy, there is a risk of venous thromboembolism, and potentially cardiovascular disease and cancer [10].

In clinical settings, gender affirmation is a complex and long-lasting procedure, involving many different healthcare professionals including psychiatrists, psychologists, endocrinologists, plastic surgeons, speech and language therapists, and counsellors. This procedure also needs a multidisciplinary approach with a schedule that allows time for physical and social transition [11,12,13]. The World Professional Association for Transgender Health (WPATH) published the most commonly accepted clinical guidelines and ‘Standards of Care’ for TGNC [14], including established general eligibility criteria for feminising or masculinising hormone therapy. These include (a) persistent, well-documented gender dysphoria; (b) capacity to make a fully informed decision and to consent for treatment; (c) age of majority in a given country; (d) if significant medical or mental concerns are present, they must be reasonably well-controlled.

The aim of this article is to raise awareness among mental health professionals about a phenomenon that is already known in specialist settings (e.g., gender affirmation clinics and substance misuse services) but less well known in different settings. Central to this paper is a collection of different clinical cases collected in inpatients and outpatients National Health Service (NHS) clinics in Suffolk (a rural county north east of London) and London (Camden and Islington Boroughs). It is also desirable that by raising awareness, TGNC patients will find that mental health clinicians are not only therapists but also advocates and allied in their gender affirmation journey. In response to the existing lack of knowledge among the health professionals, NHS England recently released [15] a document assessing the individual suitability for endocrine and other pharmacological treatments. Suggested arrangements for medical practitioners include: (a) prescription of endocrine and other pharmacological interventions for the purpose of harm reduction and acting in the best interest for reducing gender dysphoria; (b) the assessment of risks, benefits and limitations of such a pharmacological intervention and the assurance that the individual meets the relevant eligibility criteria set out by the World Professional Association for Transgender Health Standards of Care (2011); (c) the provision of patient-specific prescribing guidance to the General Practitioner (GP), including adequately-detailed information about the necessary pre-treatment assessments, and advice on dosages, administration, initiation, duration of treatment among others; (d) the preparation of written advice to the GP when the individual is discharged. Further details on specific treatments have been outlined in Table 1. An additional statement by the General Medical Council (GMC) clarifies that GPs can prescribe hormones to TGNC individuals [16].

Despite the availability of such clinical guidance and advice, TGNC individuals still do not receive the care they often need because of stigma, discrimination, and lack of awareness in health care settings [17,18]. A large population study from the 2014-5 Behavioural Risk Factor Surveillance System by Gonzales et al. found that “TGNC adults were more likely to be uninsured and have unmet health care needs and were less likely to have routine care, compared to cisgender (non-transgender) women”. Reasons for such barriers to health care included discrimination in health care, health insurance policies, employment and inadequate health policy and regulations [19]. Although very few population surveys of this kind have been carried out, another study in Ontario confirmed that 43.9% of TGNC individuals experienced inequalities in the access to healthcare and remained medically unsupervised [20].

An additional element of concern to this phenomenon is the self-administration of CHT without clinical supervision. Evidence of such hazardous behaviour emerged from studies among TGNC population in Canada [21] and from patients attending gender reassignment clinics in the United Kingdom. In the latter case, it has been estimated that 23% of individuals referred to gender reassignment clinic were self-administering hormones, mainly bought online (70%). Such behaviour appeared to be more common among trans women, as 32% of the female sample was using hormones at the moment of referral. Alarmingly, individuals that were purchasing hormones online appeared to be less informed about the risks and the side effects of hormonal therapy [22].

The growing number of illicit online pharmacies selling counterfeited products, including “Performance and Image-Enhancing Drugs” (PIEDs) and sexual enhancers [23] taken to enhance human abilities in a myriad of spheres, is another important emerging faucet within this [24,25,26,27]. PIEDs include substances with a perceived ability to enhance physical performance, psychological status and appearance, cognitive abilities and social relations, and as such, are sometimes referred to as ‘lifestyle drugs’. It has been estimated that approximately 97% of websites selling pharmaceutical products are of illicit nature [26]. Individuals can purchase a wide range of unregulated and untested medicines in these websites which are freely sold without a prescription, and at discounted prices [27,28].

## 2. Clinical Cases

A number of TGNC patients cases that have started the gender affirmation process without any medical/specialist support were collected in two general adult psychiatry assessment clinics (outpatients) in London and Suffolk between 2014 and 2018. Information was obtained as part of the routine history taking during a psychiatric assessment and no specific questionnaire was designed for the psychiatry interview. The patients were informed at the moment of the assessment that some of the information given in an anonymised version would be used for a case presentation and a case report article. The patients that were included in the article were requested to give informed consent. The assessment clinics are the first point of contact with mental health where new referrals from GPs and other health professionals are seen. Suffolk is a rural county, predominantly a white English population, with chronic lack of access to mental health and other specialist health services that are conversely available in London, where most of the specialist and national health services are located. Consulted patients were either attending the clinics, or inpatient in psychiatric wards for the assessment and treatment of mental health conditions unrelated to their gender definition. The clinics and the wards were for general adult patients only. During the psychiatric interview, it emerged, worryingly that a certain number of TGNC individuals were purchasing hormones through illegal on-line pharmacies and were using them without any medical advice or monitoring (not even at General Practitioner level) using CHT protocols that were available online or receiving advice from online forums and blogs.

The length of the current clinical procedure, which involves long waiting lists, various passages of assessment and treatment, was criticized and perceived as a barrier for receiving the standard treatment. Patients preferred to purchase the hormones online and advocated a quick and easy CHT while trusting unsolicited online protocols from non-medical professionals for a faster result. Mistrust of medical professionals has also been previously reported as a potential cause [29]. Resilience on the Internet for medical advice concerning injecting practices and dosages among other features also indicates an underlying lack of engagement with medical professionals and limited practitioner knowledge regarding these patterns of use [30].

Hormone therapy in gender affirmation may affect different organs and systems [31]. For this reason, any hormonal treatment should be prescribed and supervised by a specialist and should also be discussed in depth with the patient to prepare him/her for the treatment. In this way, it is possible to monitor and manage the treatment effectively as well as to address any of the side effects described above. Hormonal treatment with oestrogens also requires diuretics to counteract the water retention associated with their use. Diuretics should also be used under medical supervision, and the renal function and electrolytes of the patients should be checked regularly to prevent, especially in summer, potentially dangerous dehydration and electrolytes imbalances.

In this paper, seven cases of TGNC individuals are described. They were assessed in a psychiatric clinic or admitted onto a psychiatric ward for different reasons, and they were using hormones and other drugs for gender affirmation without any medical supervision and purchasing all these medications through unlicensed online dealers.

### 2.1. Oestrogens

Case 1 is a 24-year-old trans woman, working as a plumber, and single. She was referred for a psychiatric screening as the first step for the referral pathway to the Gender Identity Clinic after she disclosed to her GP that she was using hormones purchased online. The patient has no previous history of mental or physical illness, she described herself as TGNC since age 14 and started the transition, without medical supervision two years before the assessment. The patient joined on line forums where she received the information regarding the hormonal protocols and the websites selling hormones. She started the treatment on her own and subsequently asked the GP to be referred for gender reassignment/gender affirmation. At the time of the assessment, the patient did not present any comorbid psychiatric or physical conditions.

Case 2 is a 22-year-old trans woman with a previous history of depression at the age of 14 that was successfully treated with a Serotonin Reuptake Inhibitor (SSRI). The patient stated that she felt like a person “trapped in the wrong body” ever since she could remember. She complained of being bullied at school for this reason, and she thinks that the bullying and the non-acceptance from her friends caused the depressive episodes in her teens. She started the protocol online and asked the GP to continue prescribing, and the GP asked for a psychiatric opinion before proceeding. She had been using oestrogens, finasteride, and spironolactone intermittently in the last two years. The patient also has type I diabetes, treated with insulin. According to the GPs notes, the compliance with the insulin treatment and the control of his diabetes is not optimal. The mental state examination was unremarkable; the patient presented, however, traits of emotionally unstable personality disorder.

Case 3 is a 19-year-old trans woman, a college student, and single. She was referred by the GP for a psychiatric assessment following numerous suicidal attempts and self-harm episodes. She had a provisional psychiatric diagnosis of Emotionally Unstable Personality Disorder—Borderline type. During the psychiatric assessment, she disclosed using oestrogens and finasteride purchased online in order to proceed with the transition. She also reported that since she started using these hormones the emotional instability became more severe and was partly responsible for the deterioration of her clinical presentation.

Case 4 is a 26-year-old trans woman, living with a partner and working as an administrator. She was referred by her GP for psychiatric assessment as part of the procedure for a referral to the Gender Identity Clinic. The patient had no previous history of mental illness, no medical comorbidity. During the interview she disclosed using oestrogen cream and finasteride tablets purchased online although she did not disclose it before (even with the GP). The mental state examination was unremarkable.

Case 5 is a 36-year-old TGNC woman, single and unemployed. She was referred by the GP for a psychiatric assessment due to low mood and polysubstance misuse (cocaine, cannabis, amphetamine and gamma-Hidroxybutyric acid (GHB). During the interview, the patient stated that she has been taking oestrogen, spironolactone and finasteride for 15 years. A year ago, she was referred to the gender reassignment clinic where she was diagnosed with moderate depression and generalised anxiety disorder, but she was asked to see a general adult psychiatrist for the treatment. The patient complained that the main contributing factors to her depression were the stigmatisation and the lack of acceptance, in which she felt that she was victimised mainly by members of her family and her community. She also was frustrated with the length of the waiting list.

### 2.2. Androgens

Case 6 is a 25-year-old TGNC man, unemployed, living with his partner. The patient was referred for a psychiatric assessment after he disclosed to the GP about using androgens without medical advice and supervision. The patient complained of The Gender Identity Research and Education Society feeling “uneasy with his body” since the age of 12. He started to purchase androgens online, 2 years before the psychiatric assessment, following a protocol available online. The patient also has stage 4 renal failure, and he was under the care of the renal team.

Case 7 is a 27-years-old TGNC man and a university student. He was admitted to an inpatient psychiatric unit after a suicidal attempt. The patient was presenting with moderate depression, on admission, he reported of using androgens purchased online while on the waiting list for an assessment by the Gender Reassignment Clinic. The patient was using the hormones without any medical advice or supervision. The low mood and the suicidal attempt were linked to his stressful situation at the university (where he struggled to cope with academic pressure) rather than to the hormonal treatment, that he also stated that he found the hormones to be beneficial. He was also frustrated by the long waiting list before he was able to start his gender affirmation process.

## 3. Discussion and Conclusions

In our paper, we discussed the clinical cases collected by the same clinician in his clinical practice in different settings, over the span of 4 years. We are aware that it is a very limited picture and we do not think that it is necessarily representative of the entire United Kingdom. We think, however, that this raises the question of access to gender affirmation treatment and the role of every clinician as an advocate for our patients. We are aware that this phenomenon has been described before, but we believe that many clinicians are not aware of it.

Self-prescribing of sexual hormones is a widespread, but poorly studied phenomenon. As highlighted in our work, the lack of access to specialised centres, stigmatisation and marginalisation of the TGNC population as well as the motivations underlying DIY hormonal treatment, deserve further consideration.

To the best of our knowledge, this is the first report of DIY hormonal treatment in general adult psychiatric settings. Previous articles that have described the trend of self-prescribing and administrations of hormones came from sexual health clinics and gender affirmation clinics [19,20]. Psychiatric assessment clinics and psychiatric inpatient wards are often the first port of call of individuals in distress. It is very important, therefore, that the staff working in these services are aware of the particular needs of the TGNC individuals. In particular, the need to establish an environment of respect and non-stigmatisation is very important in developing an effective therapeutic relationship. It is important that the terminology used is appropriate and respectful. This applies particularly to the pronouns and it is always important to check with the patients which pronouns they prefer. It is also important to tactfully ask if they have started the gender affirmation process on their own without any clinical supervision. Psychiatrists, GPs, Sexually Transmitted Diseases’ (STD) specialists and all other clinicians should be informed about this under reported trend while encouraging the safe prescribing practice of sexual hormones.

As suggested by NHS England [15], the current assessment needs to be improved by proactively asking TGNC patients whether they are taking hormones and where they are sourcing them from.

The role of mental health services is particularly important because, before the gender affirmation process, TGNC individuals suffer from a higher rate of mental illnesses and mental discomfort (often due to stigma, discrimination and non-acceptance by family and society). For this reason, mental health professionals are more likely to encounter TGNC individuals in need of support but also have a crucial role to play as an advocate. It is, therefore, important when assessing TGNC individuals to respectfully enquire whether they have started the transition and if this is happening under medical care or not. If not, it is necessary to ask if they are sourcing hormones and other medications online or from other unlicensed sources. If this is the case, clinicians have the duty not only to inform the patients of the risk but also to suggest safer alternatives and support if necessary, the individuals in this process.

A common theme that emerged from our case studies was the use of hormones purchased online without any clinical guidance or supervision before and during the treatment. As previously argued [28,32], the Internet often provides a channel for accessing peer-group experiences and disseminating such risky behaviours. The underlying interpersonal trust embedded in such sub-cultural groups, as seen for instance on discussion fora or social networking, can reinforce the establishment of risk taking norms, especially among early adopters. In addition, the intake of previously untested and unregulated medicinal products can expose users to a series of unwanted side-effects, especially in potentially risky, if unsupervised, medical practices such as intramuscular (IM) injections [33,34]. The shipping process is also questionable with the risk that even where the product is genuine, it may arrive in a condition that renders it unsafe for use. Buyers may also receive counterfeited products and, therefore, using compounds that may be toxic or even lethal. It is also concerning that substances like hormones which have a significant effect on the body and the mind, are used without guidance and monitoring of the side effects. Furthermore, the route of administration (e.g., IM) can lead to additional health risks, both chronic and acute [33].

Further studies need to be carried out to evaluate the motivation underlying such a poorly researched trend. The main reason for such behaviours seems to be due to the difficulty in accessing gender reassignment/gender affirmation treatments, leaving a significant part of the TGNC population in a condition of discomfort that make them more vulnerable to the onset of psychiatric illnesses (depression, anxiety) as well as other unhealthy conditions or practices (e.g., smoking, drug intake) [18,35,36]. Furthermore, as their need for medical assistance may grow [36], trans individuals often experience problematic accesses to healthcare in terms of professionals’ education and discriminatory practices [37]. This is also due to the length of the waiting lists (up to three years) that pushes TGNC individuals into obtaining hormones online [38]. Despite the various efforts made by NHS England, the GMC and the Royal Colleges to improve the situation, a significant group of individuals still do not receive the care that they need and deserve in the United Kingdom. This may also be due to the chronic lack of funds for these services despite recent years’ additional funding [39]. Gender affirmation is not the treatment of an illness but is a procedure that improves the wellbeing of TGNC and non-binary individuals and, therefore, should be appropriately funded and supported by every clinician [40].

It is necessary, therefore, to address such gaps in public health policy and clinical practice knowledge regarding gender affirmation and establish alternative services. Local clinic services, complementary to the NHS provided treatment, may be crucial to the wellbeing of individuals who are feeling disenfranchised and are not attending the gender reassignment/gender affirmation clinics while considering the treatment [41]. Examples of such services can be found in the United States with the Transgender Health Services program (STRIDE) based in San Francisco, which is structured as a peer-based model providing hormone therapy and support for the general and the psychological health of TGNC individuals [42]. Different gender reassignment/gender affirmation programs are also available in Canada. In British Columbia and Ontario projects like Trans Care BC (British Columbia) and Trans Health Connection (Ontario) aim to facilitate the access to care for TGNC by providing information and support for the TGNC community [43,44].

In summary, we believe that increasing the access to gender affirmation services alone, however, is not enough and it is necessary to raise awareness among every clinician about the special needs that TGNC individuals have when accessing healthcare. It would be necessary, to achieve this goal, to design and disseminate among all clinicians, a questionnaire regarding their attitude towards TGNC individuals, their knowledge about the gender affirmation process and the specific clinical needs that the TGNC population has. In this way, it would be possible to establish the educational needs that clinicians have and, therefore, consider specific training. This would enable improvement of access to care for TGNC individuals by fighting stigma and creating a more inclusive service. This approach should be across all the services that might care for TGNC individuals. Furthermore, disseminating knowledge and raising awareness might have another beneficial effect by transforming conscious clinicians in health to advocate for the TGNC population. In this way, it may be possible to address not only the problem highlighted in this article (the difficulty of accessing gender affirmation clinics) but also the more widespread discrimination that TGNC individuals face when accessing various health care services.

As we have seen in our study, the implementation of such different and innovative health care services for TGNC individuals as well as more targeted prevention strategies on such underreported and highly risky behaviours have become a necessity in the United Kingdom and elsewhere. Closer attention should also be paid to the online market of DIY hormones, and an open dialogue with LGBTQ organisations should be established to support TGNC individuals and better understand their unmet needs.

## Figures and Tables

**Table 1 brainsci-08-00088-t001:** Types of treatment for masculinisation and feminisation [15]

Aim	Type of Preparation	Notes	Recommendations
Medications for masculinisation	Testosterone preparations	Include testosterone injections and transdermal gels	Avoid smoking (risk of thrombosis)
	Medications to suppress hyptolamic-pituitary-gonadal activity		Avoid smoking (risk of thrombosis)
Medications for feminisation	Estradiol preparations	Doses necessary to achieve serum estradiol levels typical of pre-menopausal woman. Include oral estradiol and transdermal estradiol as patches and gels (for people over 40 years old). Ethinylestradiol will not be recommended	Avoid if Body Mass Index > 40; avoid smoking (risk of thrombosis)
	Medications to suppress hypothalamic-pituitary-gonadal activity and endogenous testosterone release	Include gonadotropin releasing hormone analogues and 5-alpha reductase inhibitors	Avoid if Body Mass Index > 40; avoid smoking (risk of thrombosis)
	Ornithine decarboxylase inhibitors	May be recommended as an adjunct to facial hair reduction interventions	Avoid smoking (risk of thrombosis)

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
