# Peer review of "Transitioning Bodies. The Case of Self-Prescribing Sexual Hormones in Gender Affirmation in Individuals Attending Psychiatric Services"

_brainsci, 2018, doi:10.3390/brainsci8050088_

Round 1

Reviewer 1 Report

The authors explain in their comments that the paper is novel as it looks at people attending psychiatric clinics. I don’t feel that this is clear enough in the actual paper and therefore the novelty of the paper is not clear. The fact that this group was explored should be mentioned in the title as well as being discussed in the introduction.

The authors mention that they have added a brief description of the clinics and the geography. I couldn’t find this in the introduction (as this is the section I was commenting on). Throughout your responses it would have been helpful to say where (line and page number) you addressed each comment.

The authors mention they have made it clearer as to why the article is needed. This could be emphasised further (see above comment).

There are several comments that you have not responded to towards the end?

Author Response

The authors explain in their comments that the paper is novel as it looks at people attending psychiatric clinics. I don’t feel that this is clear enough in the actual paper and therefore the novelty of the paper is not clear. The fact that this group was explored should be mentioned in the title as well as being discussed in the introduction.

 We have added a comment in the introduction and also I have modified the title accordingly

The authors mention that they have added a brief description of the clinics and the geography. I couldn’t find this in the introduction (as this is the section I was commenting on). Throughout your responses it would have been helpful to say where (line and page number) you addressed each comment.

Following your suggestion we have clarified also in the introduction

The authors mention they have made it clearer as to why the article is needed. This could be emphasised further (see above comment). 

We have also emphasized further as you suggested

Reviewer 2 Report

This manuscript presents reflections on an important phenomenon concerning transgender population, or rather the self-prescription of sexual hormones. Authors collected some information from seven transgender individuals, reporting that the main reason for this behavior is the perceived discrimination within healthcare clinics and long waiting lists. The intent of the authors is laudable, above all because they wanted to address a still little explored phenomenon. That being said, there are serious limitations that should be addressed. The most important limitation concerns a methodological issue. It is not clear how authors deduced their conclusions, for instance that transgender people self-prescribing hormones feel discriminated by healthcare professionals. A methodology section is totally missing and it is not possible to evaluate the strength of authors’ conclusions. Authors should provide data about percentages of trans people not self-prescribing hormones and trans people self-prescribing hormones in their clinics and compare the levels of perceived discrimination to be sure that this dimension (i.e., discrimination) affects such a behavior. On the contrary, if authors do not have these data, they should qualitatively analyze collected narratives of trans people self-prescribing hormones, through a clear theoretical framework (e.g., minority stress theory) and an adequate methodology (e.g., interpretative phenomenological analysis, constant comparison analysis, or other). Unfortunately, it is hard for me to understand what follows the clinical cases, because it should derive from the data analysis, on which authors should do much work.

I hope that these suggestions might help authors to make their manuscript stronger. Below, authors will find some other little suggestions concerning the introduction paragraph:

1.      The sentence “A significant proportion of the population define themselves as transgender, intersex non-binary or gender non-conforming” (lines 32-33) requires a specification. How many transgender, intersex non-binary or gender non conforming people exist?

2.      The same is for the sentence “A significant number of these individuals often decide to undergo a gender affirmation process”. How many trans people decide to undergo surgery? Today, it is really different compared to the past and we are witnessing a decline of the gender affirming surgery. Authors should problematize this point.

3.      Authors use too many different terms to refer to transgender population (e.g., trans, transgender, transsexual, etc.). I suggest to use the term “transgender and gender nonconforming people” use the abbreviation “TGNC” throughout the text, in according to the most recent APA guidelines (APA, 2015). On the contrary, a non-expert reader might be confused.

Author Response

This manuscript presents reflections on an important phenomenon concerning transgender population, or rather the self-prescription of sexual hormones. Authors collected some information from seven transgender individuals, reporting that the main reason for this behavior is the perceived discrimination within healthcare clinics and long waiting lists. The intent of the authors is laudable, above all because they wanted to address a still little explored phenomenon. That being said, there are serious limitations that should be addressed. The most important limitation concerns a methodological issue. It is not clear how authors deduced their conclusions, for instance that transgender people self-prescribing hormones feel discriminated by healthcare professionals. A methodology section is totally missing and it is not possible to evaluate the strength of authors’ conclusions. Authors should provide data about percentages of trans people not self-prescribing hormones and trans people self-prescribing hormones in their clinics and compare the levels of perceived discrimination to be sure that this dimension (i.e., discrimination) affects such a behavior. On the contrary, if authors do not have these data, they should qualitatively analyze collected narratives of trans people self-prescribing hormones, through a clear theoretical framework (e.g., minority stress theory) and an adequate methodology (e.g., interpretative phenomenological analysis, constant comparison analysis, or other). Unfortunately, it is hard for me to understand what follows the clinical cases, because it should derive from the data analysis, on which authors should do much work. 

We fully appreciate the suggestions from the reviwes. Our contrbution is to present a selected number of case-studies and to raise awarness about the phenomenon and its potential harms rather than a cross-sectional analysis. As a result of this, a further and more arcticlated work will be planned. 

I hope that these suggestions might help authors to make their manuscript stronger. Below, authors will find some other little suggestions concerning the introduction paragraph:

1.      The sentence “A significant proportion of the population define themselves as transgender, intersex non-binary or gender non-conforming” (lines 32-33) requires a specification. How many transgender, intersex non-binary or gender non conforming people exist? 

We have added, as suggested some statistic data from the USA and the UK. 

2.      The same is for the sentence “A significant number of these individuals often decide to undergo a gender affirmation process”. How many trans people decide to undergo surgery? Today, it is really different compared to the past and we are witnessing a decline of the gender affirming surgery. Authors should problematize this point. 

We added the estimate for the UK however we could not find in the literature the trend you describe 

3.      Authors use too many different terms to refer to transgender population (e.g., trans, transgender, transsexual, etc.). I suggest to use the term “transgender and gender nonconforming people” use the abbreviation “TGNC” throughout the text, in according to the most recent APA guidelines (APA, 2015). On the contrary, a non-expert reader might be confused.

We have changed the terminology as suggested. We have left other terms like transgender only when we have been quoting verbatim a reference.

Round 2

Reviewer 2 Report

I recommend publication in the current form. I feel that authors have clarified my doubts.